# BERTie Bott's Every Flavor Labels: A Tasty Introduction to Semantic Role Labeling for Galician

**Micaella Bruton**
Uppsala University
micaella.bruton.4363@student.uu.se

**Meriem Beloucif**
Uppsala University
meriem.beloucif@lingfil.uu.se

## Abstract

In this paper, we leverage existing resources, such as WordNet and dependency parsing, to build the first Galician dataset for training semantic role labeling systems in an effort to expand available NLP resources. Additionally, we introduce Verbal Indexing, a new pre-processing method, which helps increase the performance when semantically parsing highly complex sentences. We use transfer learning to test both the resource and the Verbal Indexing method. Our results show that the effects of Verbal Indexing were amplified in scenarios where the model was both pre-trained and fine-tuned on datasets utilizing the method, but improvements are also noticeable when only used during fine-tuning. The best-performing Galician SRL model achieved an *f1* score of 0.74, introducing a baseline for future Galician SRL systems. We also tested our method on Spanish where we achieved an *f1* score of 0.83, outperforming the baseline set by the 2009 CoNLL Shared Task by 0.025, showing the merits of our Verbal Indexing method for pre-processing.

## 1 Introduction

The goal of semantic role labeling (SRL), also known as shallow semantic parsing, is to define the semantic roles of any given word in a sentence in regard to a specific noun or verb; the focus of this study specifically is SRL in regard to verbal predicates (Jurafsky and Martin, 2023). Focus is placed on directly mapping a predicate to its role and recipients (Màrquez et al., 2008). Computational systems are then able to use these roles to establish a shallow semantic representation of a string of words, allowing for inferences which otherwise would be lost to the system (Jurafsky and Martin, 2023). As these roles seek to remove ambiguity as much as possible, they are specific to each verb's sense meaning, though tendencies can be identified. In general, *Arg0* represents that which causes an event or a change of state in another

participant with respect to the verbal root, while *Arg1* represents that which is undergoing a change of state or causally affected by another participant (Jurafsky and Martin, 2023).

| | arg0 | root | | arg1 | |
|---|---|---|---|---|---|
| **glg** | Touriño | dá | un | xiro | ... |
| **eng** | Touriño | takes | a | turn | ... |

Table 1: Introduction to SRL: Excerpt from Galician Dataset Sentence #66
‖ *glg* – Galician text | *eng* – English text | *arg0* – token assigned as Argument 0 | *root* – token assigned as the verbal root | *arg1* – token assigned as Argument 1 ‖

Table 1 contains an excerpt from the Galician dataset introduced in this study alongside an English translation to showcase the identification of roles, or arguments, within a sentence. As can be seen in the table, the main verbal root in this excerpt is *dá* or *takes*, *Touriño* is identified as *Arg0* as they are performing the action described by the verb, and *xiro* or *turn* is identified as *Arg1* as it is that which is being affected by *Arg0*.

Unfortunately, corpora including this shallow semantic information are extremely few and far between, with .1% of the languages of the world having the necessary language-specific information readily available and only .3% having access to the translated version of English PropBank, the most extensive SRL resource currently available (Eberhard et al., 2023; Palmer et al., 2005; Jindal et al., 2022; Akbik et al., 2015). This lack of data makes it nearly impossible for SRL systems to be developed for the vast majority of languages.

The main purpose of this work was to produce a dataset and initial baseline for Galician SRL, a low-resource Western Ibero-Romance language with approximately 4 million speakers worldwide (Britannica, 2023). Despite its large number of speakers, NLP tools and resources for Galician are extremely limited and it remains classified as a low-resource

| Tokens (eng) | Touriño | takes | a | turn | in | | his | campaign strategy | | |
|---|---|---|---|---|---|---|---|---|---|---|
| Tokens (glg) | Touriño | dá | un | xiro | en | a | súa | estratexia | de | campaña |
| ID Tags | 1 | 4 | 0 | 2 | 0 | 0 | 0 | 0 | 0 | 0 |
| Roles | r0:arg0 | r0:root | O | r0:arg1 | O | O | O | O | O | O |

Table 2: SRL of Galician Dataset Sentence #66
‖ *eng* – English text | *glg* – Galician text | *r[idx]:[label]* – link [label] to [idx]th verb | *root* – token assigned as the verbal root | *arg0* – token assigned as argument 0 | *arg1* – token assigned as argument 1 | *O* – token assigned as not directly related to root verb‖

language for most NLP tasks; it is our hope that the release of this dataset will help to improve and expand resources for Galician (Language Archives Services, University of British Columbia, 2023; Agerri et al., 2018).

Several transfer-learning experiments were also performed to determine the efficacy of leveraging existing high-resource SRL systems in the development of languages with lesser amounts of data available as it has seen success previously (Okamura et al., 2018; Alimova et al., 2020; Oliveira et al., 2021; Daza Arévalo, 2022).

## 2 Related Work

Recently, methods such as Descriptive Semantic Role Labeling (DSRL) and the inclusion of syntactic information have been proposed in an effort to increase interpretability, flexibility, and overall performance of SRL systems (He et al., 2018; Conia et al., 2022). DSRL seeks to shift away from traditional argument labeling in favor of labeling through descriptions using natural language (Conia et al., 2022). Both methods have seen great success, performing on par with or beating current state-of-the-art results in high-resource contexts for English and Chinese (He et al., 2018; Conia et al., 2022). Despite their success, a choice to utilize standard methods of argument labeling was made due to the extremely low-resource status held by Galician and the overall current lack of resources.

## 3 Method

Development and training scripts can be found on the project GitHub[1] while the fully trained models and datasets can be accessed via HuggingFace[2].

### 3.0.1 Galician

While Galician does have a publically available WordNet and syntactically-parsed corpora, no SRL

[1]https://github.com/mbruton0426/GalicianSRL
[2]https://huggingface.co/mbruton

corpora exist outside of that produced by this work. Syntactic information contained in two of these resources, the UD Galician-CTG and UD Galician-TreeGal treebanks, were leveraged to create a simple SRL dataset for Galician (Gómez-Rodríguez et al., 2017; Alonso et al., 2016). The UD Galician-CTG treebank contains technical sentences of the medical, sociological, ecological, economical, and legal domains that have been pulled from the Galician Technical Corpus and automatically parsed by the TALG NLP Research Group at the University of Vigo (Gómez-Rodríguez et al., 2017). The UD Galician-TreeGal treebank was developed at the LyS Group at the Universidade de Coruña and was derived from a subset of the XIADA corpus created at the Centro Ramón Piñeiro para a Investigación en Humanidades (Alonso et al., 2016).

These datasets were downloaded from their respective project GitHub in CoNLL-U format and processed via a Python script. For each unique sentence in these corpora, the total number of root verbs were assigned an index which was then used to match arguments to their respective verb. Using the NLTK library for Galician WordNet, a synset for each verbal lemma was identified and matched to its English PropBank role set (Bird et al., 2009; Guinovart and Portela, 2018; Palmer et al., 2005). This information was then used to assign one of five roles to each token: "r[idx]:root", "r[idx]:arg0", "r[idx]:arg1", "r[idx]:arg2" or "O"; where *idx* designates the verb index and *O* designates a non-involved token. An attempt to identify additional arguments or non-verbal roots was not made at this time, and verbal lemmas unable to be linked to a single, specific English PropBank role were not included.

Table 2 gives an example of a sentence with one root verb and its arguments; for each additional root appearing in a sentence, the index number will increase by one. Figure 1 showcases how decisions were made throughout the automated argument classification process; Upon completion, 25%

of sentences were randomly selected for manual validation. In cases where a clear distinction was not able to be made, the sentence was removed from the dataset. The final dataset contains 3,987 training sentences and 998 test sentences. Sentences in the dataset include up to 13 verbal roots, with arguments for each root varying between arg0, arg1, and arg2.

### 3.0.2 Spanish

The 2009 CoNLL Shared Task Spanish data was used to develop the Spanish SRL dataset; it contains 14,329 training sentences, 1,655 development sentences, and 1,725 test sentences (Hajič et al., 2009). Sentences were originally extracted from the AnCora-ES corpus, developed by the Centre de Llenguatge i Computació de la Universitat de Barcelona which were taken from newspaper articles (Rodríguez et al., 2004). Pre-processing of the data was done to implement Verbal Indexing; additional thematic information included in the Spanish data was preserved. Sentences in the dataset include up to 16 verbal roots and the following role labels: "r[idx]:root", "r[idx]:arg0 | [*agt, cau, exp, src*]", "r[idx]:arg1 | [*ext, loc, pat, tem*]", "r[idx]:arg2 | [*atr, ben, efi, exp, ext, ins, loc*]", "r[idx]:arg3 | [*ben, ein, fin, ori*]", "r[idx]:arg4 | [*des, efi*]", and "r[idx]:argM | [*adv, atr, cau, ext, fin, ins, loc, mnr, tmp*]".

### 3.1 Models

24 models in total were developed and tested, 16 Galician and 8 Spanish SRL models. To our knowledge, no SRL work on Galician has been published and our results introduce an initial baseline for future work. Due to this, Spanish models were developed not only as an additional transfer-learning scenario for Galician but also to test our method against an established baseline. Results from the 2009 CoNLL Shared Task Spanish SRL-only models are used as this baseline (Hajič et al., 2009). Pre-trained language models mBERT and XLM-R were used as main architectures and fine-tuning followed the same basic structure for all models (Devlin et al., 2019; Conneau et al., 2020). As each sentence goes into a model, it is tokenized and input all at once. The model is then expected to make predictions as to the role of each token within the sentence; additionally, the model is expected to assign each verbal root an index and link each argument to its root via this index. Specific arguments vary by language as defined above.

The Verbal Indexing method proposed here differs from basic direct mapping as it involves tallying the number of verbs present in a given sentence (Màrquez et al., 2008). This is done to enhance the ability of the model to understand the sentence's complexity and aid in the initial identification of verbs within the sentence. This method does not assign semantic roles or rulesets; the purpose of Verbal Indexing is to count all verbs within a sentence, arranging them based on their position in the sentence. Importantly, this indexing process does not impact the PropBank label associated with each verb. For instance, even if a verb with the PropBank label *run.01* appears twice in a single sentence, each instance would be assigned a distinct index through the Verbal Indexing approach, as the index is directly linked to the verb's position in the sentence.

Monolingual models were fine-tuned for the SRL task in the target language on the base version of the chosen language model. Transfer-learning models were previously trained on the SRL task in either English, Portuguese, Spanish (for Galician) or some combination of these, then fine-tuned for the SRL task in the target language. Models are named using the following format, [*target-SRL-language*]_[*pretrained-SRL-language*]_[*base-language-model*], where the target language is identified by a 3-letter identifier, the pre-trained language(s) are identified by a two-letter identifier, and the base language model is identified by its name.

To reduce computational and environmental costs, English and Portuguese pre-trained SRL models were adopted from HuggingFace and used as published by Oliveira et al. (2021). English models were pre-trained using the OntoNotes v5.0 English SRL corpus, including 316.155 sentences from news, conversational telephone speech, weblogs, Usenet newsgroups, broadcast, and talk shows while the Portuguese models used data from the PropBank-BR Brazilian Portuguese SRL corpus, including 13,665 sentences taken from newspapers (Duran and Aluísio, 2011; Weischedel et al., 2013; Oliveira et al., 2021). As these models were used as-is, they were not fine-tuned using our Verbal Indexing method, but rather IOB-sequencing in the format [I,O,B]-[label] (Oliveira et al., 2021).

After models trained to convergence, they were evaluated on their overall performance via *f1* score; their overall ability to identify verbal roots and their

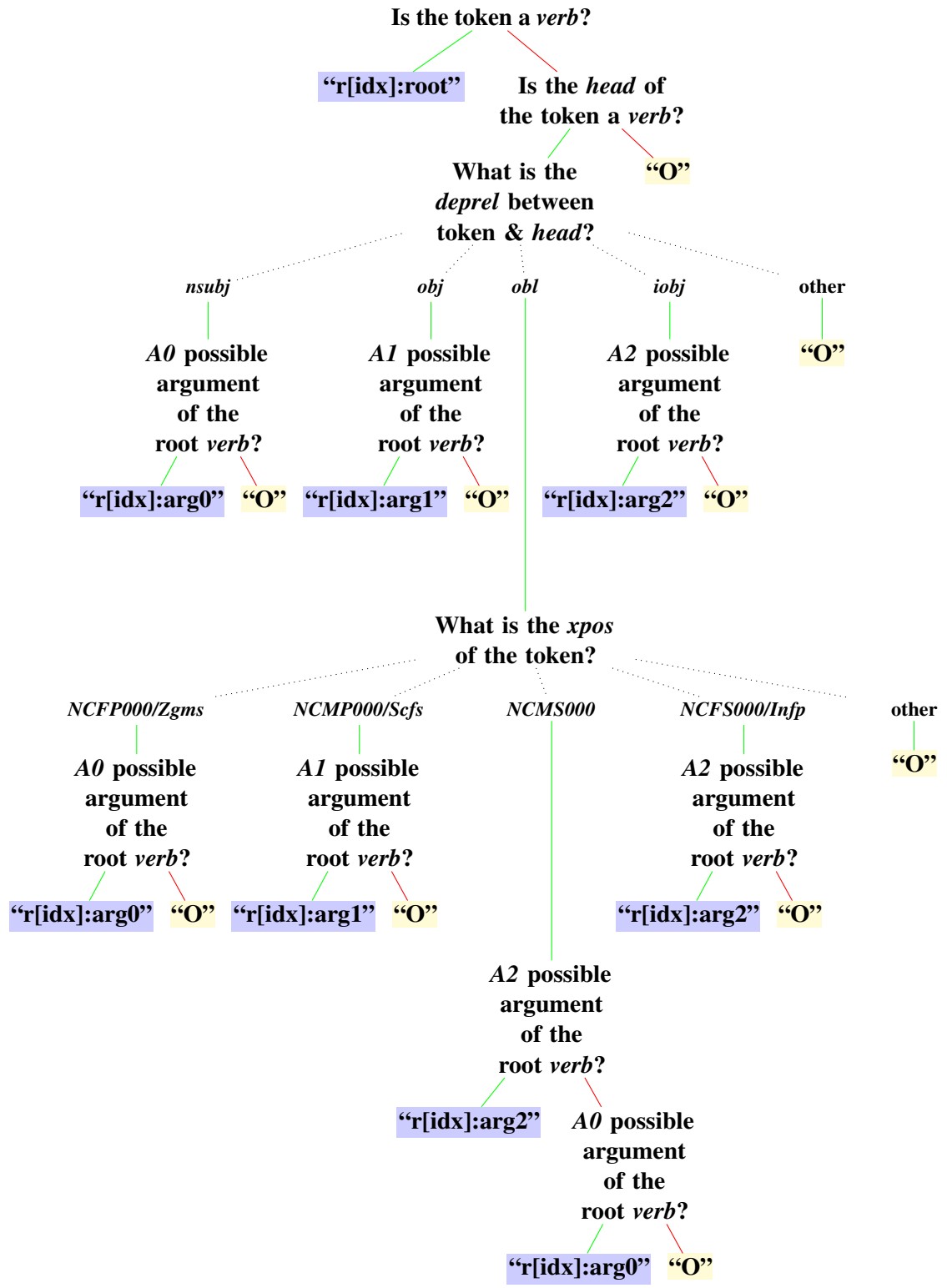

Figure 1: Argument Classification Decision Tree

|| green line – affirmative decision | red line – negative decision | *r[idx]:[label]* – format of label assigned to arguments, link [label] to [idx]th verbal root | *root* – assigned as the verbal root | *arg0/A0* – assigned as argument 0 | *arg1/A1* – assigned as argument 1 | *arg2/A2* – assigned as argument 2 | *O* – assigned as not directly related to root | *deprel* – dependency relation | *nsubj* – nominal subject | *obj* – object | *obl* – oblique | *iobj* – indirect object | *xpos* – language specific part-of-speech tag ||

arguments via *f1* score; and their ability to index verbal roots and match them to their respective arguments via *f1* score, using the Python framework seqeval (Nakayama, 2018). Scores are reported on a scale of 0.0-1.0.

# 4 Results

Results are summarized across four tables, two for each language. Overall *f1* score and score by label are presented for Galician models in Table 3 and Spanish models in Table 6. *f1* score by Verbal Indexing set are shown in Table 4 for Galician and 7 for Spanish. Verbal Indexing sets are identified in the table as "r[idx]", where [idx] includes the average *f1* score for the identification of the verbal root and all arguments of the given index.

## 4.1 Galician Models

In general, XLM-R models outperform mBERT models; even considering just the monolingual models, gal_XLM-R model outperforms gal_mBERT model by 0.02. Transfer-learning models pre-trained on Spanish tend to perform better, with the XLM-R model pre-trained on English, then Spanish, gal_ensp_XLM-R, performing best overall; achieving a score of 0.74 for a total improvement of +0.06 over the worse performing monolingual model. Despite its top overall score, it slightly underperforms in the identification of verbal roots, arg0 and arg1; the XLM-R model pre-trained on Portuguese, then Spanish, gal_ptsp_XLM-R, achieves the top score for identifying verbal roots and arg1 and the XLM-R model pre-trained solely on Spanish, gal_sp_XLM-R, achieves the top score for identifying arg0.

When considering identification by root set however, the top performing model, gal_ensp_XLM-R, scores just 0.01 under gal_ptsp_XLM-R at 0.79 when identifying the root and arguments for 0th root but maintains the top score for roots and arguments for the 1st through 5th sets; it is also the only model able to identify any portion of the 9th root set. In fact, all models not pre-trained on the Verbal Indexing method were unable to identify anything past the 6th root set.

## 4.2 Spanish Models

Again XLM-R models outperform mBERT, however, differences in scores only range from 0.01-0.02. Transfer-learning XLM-R models also outperform the monolingual models, but only by 0.01;

these models all tie for best performing model overall with a score of 0.83. spa_mBERT receives the top score of 0.51 when identifying roots and ties with the pre-trained Portuguese XLM-R model, spa_pt_XLM-R, with a top score of 0.39 when identifying arg0. spa_pt_XLM-R also produces top-level results when identifying arg1, arg2, and argM with scores of 0.39, 0.36 and 0.30 respectively. The English pre-trained XLM-R model, spa_en_XLM-R, outperforms spa_pt_XLM-R in the identification of arg3 and arg4 with an increase of 0.04 and 0.07 correspondingly.

spa_en_XLM-R also receives the most top scores for Verbal Indexing, achieving scores of 0.573, 0.499, 0.469, and 0.306 for the 1st, 2nd, 3rd, and 7th root sets. The monolingual XLM-R model outperformed all other identifications of the 0th root set by 0.006 or more, but had the worst performance for most sets past r3. All models but the duel pre-trained mBERT model, spa_enpt_mBERT, were able to identify up to 9 root sets, albeit poorly. No model was able to identify any part of a root set r10 or greater. None of these models were pre-trained using the Verbal Indexing method, and as such all models stop identifying at the 9th root set.

Table 5 contains the *f1* scores for Spanish SRL models published by the 2009 CoNLL Shared Task, as well as the results from the top performing Spanish model produced by this study, listed under *Bruton*.

# 5 Discussion

Based on the performance of the monolingual models, XLM-R clearly outperforms mBERT as a base architecture in the low-resource scenario. As both architectures are pre-trained on the same, unrelated task, have the same number of layers, hidden state, and attention heads, and include Galician and Spanish unlabeled text in their pre-training languages, this performance boost appears to be influenced by the difference in tokenization methods utilized by each architecture; mBERT uses WordPiece while XLM-R uses SentencePiece.

WordPiece is an algorithm that breaks down words into sub-words using a variation of Byte-Pair Encoding (BPE) able to use a range of n-gram sizes, while SentencePiece is a re-implementation of sub-word units able to utilize both traditional BPE and unigram language modelling (Wu et al., 2016; Kudo and Richardson, 2018). Generally, WordPiece is seen to perform better in text classi-

| Model [e2c] | tot_avg | root | arg0 | arg1 | arg2 |
|---|---|---|---|---|---|
| gal_mBERT[6] | **0.68** | **0.39** | 0.32 | 0.32 | 0.30 |
| gal_en_mBERT[6] | 0.69 | 0.41 | 0.31 | 0.31 | **0.28** |
| gal_pt_mBERT[6] | **0.68** | 0.41 | 0.35 | **0.30** | **0.28** |
| gal_sp_mBERT[4] | 0.72 | 0.53 | 0.40 | 0.36 | 0.33 |
| gal_enpt_mBERT[6] | 0.70 | 0.43 | 0.33 | 0.32 | 0.31 |
| gal_ensp_mBERT[4] | 0.70 | 0.45 | 0.38 | 0.34 | 0.30 |
| gal_ptsp_mBERT[3] | 0.69 | 0.46 | 0.38 | 0.34 | 0.30 |
| gal_enptsp_mBERT[4] | 0.70 | 0.52 | 0.36 | 0.37 | 0.31 |
| gal_XLM-R[6] | 0.70 | **0.39** | **0.30** | 0.32 | 0.31 |
| gal_en_XLM-R[7] | 0.70 | 0.45 | **0.30** | 0.33 | 0.31 |
| gal_pt_XLM-R[6] | 0.70 | 0.43 | 0.33 | 0.32 | **0.28** |
| gal_sp_XLM-R[6] | 0.73 | 0.52 | **0.47** | 0.40 | 0.34 |
| gal_enpt_XLM-R[6] | 0.69 | 0.41 | 0.32 | 0.31 | 0.29 |
| gal_ensp_XLM-R[6] | **0.74** | 0.53 | 0.42 | 0.41 | **0.40** |
| gal_ptsp_XLM-R[6] | 0.73 | **0.56** | 0.41 | **0.42** | 0.38 |
| gal_enptsp_XLM-R[6] | 0.71 | 0.45 | 0.37 | 0.38 | 0.32 |

Table 3: Final *f1* Scores: Galician Models
‖ *e2c* – number of epochs trained until convergence | *tot_avg* – average *f1* score overall | *root* – average *f1* score identifying verbal roots | *arg0* – average *f1* score identifying argument 0 | *arg1* – average *f1* score identifying argument 1 | *arg2* – average *f1* score identifying argument 2 ‖

| Model [e2c] | r0 | r1 | r2 | r3 | r4 | r5 | r6 | r7 | r8 | r9 | r10 |
|---|---|---|---|---|---|---|---|---|---|---|---|
| gal_mBERT [6] | 0.78 | 0.69 | 0.62 | 0.54 | 0.50 | 0.29 | 0.11 | **0.00** | **0.00** | **0.00** | **0.00** |
| gal_en_mBERT [6] | 0.78 | 0.69 | 0.61 | **0.52** | 0.52 | **0.27** | 0.08 | **0.00** | **0.00** | **0.00** | **0.00** |
| gal_pt_mBERT [6] | 0.77 | 0.69 | **0.60** | 0.53 | 0.50 | 0.29 | 0.16 | **0.00** | **0.00** | **0.00** | **0.00** |
| gal_sp_mBERT [4] | 0.76 | 0.71 | 0.65 | 0.61 | 0.58 | 0.48 | 0.34 | 0.10 | 0.07 | **0.00** | **0.00** |
| gal_enpt_mBERT [6] | 0.77 | 0.70 | 0.62 | 0.54 | 0.51 | 0.38 | 0.15 | **0.00** | **0.00** | **0.00** | **0.00** |
| gal_ensp_mBERT [4] | 0.77 | 0.69 | 0.62 | 0.53 | 0.49 | 0.48 | 0.29 | 0.02 | **0.00** | **0.00** | **0.00** |
| gal_ptsp_mBERT [3] | **0.75** | **0.68** | 0.63 | 0.57 | 0.49 | 0.44 | 0.26 | **0.00** | 0.07 | **0.00** | **0.00** |
| gal_enptsp_mBERT [4] | 0.78 | 0.69 | 0.62 | 0.57 | 0.52 | 0.46 | 0.27 | 0.19 | 0.04 | **0.00** | **0.00** |
| gal_XLM-R [6] | **0.80** | 0.72 | 0.64 | 0.54 | 0.46 | 0.29 | **0.05** | **0.00** | **0.00** | **0.00** | **0.00** |
| gal_en_XLM-R [7] | 0.78 | 0.70 | 0.63 | 0.52 | **0.44** | 0.41 | 0.17 | 0.04 | **0.00** | **0.00** | **0.00** |
| gal_pt_XLM-R [6] | 0.79 | 0.70 | 0.63 | 0.53 | 0.51 | 0.35 | 0.09 | **0.00** | **0.00** | **0.00** | **0.00** |
| gal_sp_XLM-R [6] | 0.79 | **0.73** | 0.66 | 0.59 | 0.56 | 0.50 | 0.36 | 0.13 | **0.23** | **0.00** | **0.00** |
| gal_enpt_XLM-R [6] | 0.77 | 0.71 | 0.62 | 0.52 | 0.48 | 0.33 | 0.06 | **0.00** | **0.00** | **0.00** | **0.00** |
| gal_ensp_XLM-R [6] | 0.79 | **0.73** | **0.68** | **0.61** | **0.62** | **0.59** | 0.37 | 0.25 | 0.08 | **0.07** | **0.00** |
| gal_ptsp_XLM-R [6] | **0.80** | 0.72 | 0.65 | 0.57 | 0.58 | 0.54 | **0.42** | **0.28** | 0.13 | **0.00** | **0.00** |
| gal_enptsp_XLM-R [4] | 0.76 | 0.70 | 0.65 | 0.58 | 0.53 | 0.45 | 0.25 | 0.10 | **0.00** | **0.00** | **0.00** |

Table 4: Final *f1* Scores by Verb Index: Galician Models
‖ *e2c* – the number of epochs trained until convergence | *r[idx]* – average *f1* score identifying [idx]th root/arguments | score of 0.00 signifies that no correct identifications of any kind were made by that model for the [idx]th root ‖

| Model | f1 |
|-------|------|
| **Bruton** | **0.83** |
| **Zhao** | 0.80 |
| **Meza-Ruiz** | 0.78 |
| **Nugues** | 0.77 |
| **Baoli Li** | 0.74 |
| **Moreau** | 0.64 |
| **Täckström** | 0.62 |
| **Lin** | **0.59** |

Table 5: 2009 CoNLL Shared Task Results for Spanish SRL-only Models (Hajič et al., 2009) and top results from this study (Bruton)
|| *f1* – average *f1* scores ||

fication models and those focusing on natural language understanding and is especially successful with highly morphological languages (Wu et al., 2016; Alimova et al., 2020). SentencePiece is more suited to tasks such as sentiment analysis and named entity recognition (NER), and though is able to handle out-of-vocabulary words better than WordPiece, can struggle with morphology (Kudo and Richardson, 2018). Though Galician is a highly inflectional language and the goal of SRL is to improve natural language understanding, in our training process the labeling task was treated very much like NER, alongside indexing the verbal roots of a sentence which could explain why the XLM-R model was able to perform so much better in this scenario (Galega, 2012).

An example Galician sentence from the test dataset can be seen in Table 8 alongside the SRL results of both the gal_mBERT and gal_XLM-R. mBERT's predisposition towards morphological processing can be seen in its partial error; while the correct verb index and argument were selected, it was inaccurately applied to only a portion of the correct token. *Sexualidade* is split into two parts, *sexual* and *idade*, with only the first portion being identified as an argument. This can be seen throughout the dataset with the mBERT model; tokens are incorrectly parsed, with only a portion of the token being assigned a role, rather than the entire token. The XLM-R model manages a perfect prediction on this sentence, and generally throughout the dataset avoids mistakes of this type; its major struggles in being unable to correctly identify roots and arguments in more complex sentences; past the fourth root, its ability to identify roles drops sharply.

Despite Portuguese's similarities to Galician, models pre-trained in Spanish achieved the best results; scoring 0.04 over both the monolingual and Portuguese pre-trained models. As the Portuguese and Spanish training data are of comparable sizes and domains, the sole difference between the sets is the implementation of Verbal Indexing, suggesting this method has a positive effect on final performance.

Only minor differences could be seen between the two architectures in the higher-resource Spanish scenario, with both monolingual models achieving a score of 0.82. This indicates that the choice of base language model matters less the more target language training data exists. Pre-training on the SRL task also seemed to have little to no effect on the final *f1* score, further supporting that the most important variable toward accurate performance is the availability of data. As all scores were still able to outperform the baseline and maintained no differences in the training set other than our proposed method, this also suggests that Verbal Indexing is advantageous.

### 5.1 Identification by Verbal Indexing

When measuring the *f1* score on the identification of Galician arguments associated with the 0th, 1st, and 2nd verbal root sets, the monolingual XLM-R model was able to outperform the mBERT by 0.02, 0.03, and 0.02 respectively; root 0 tying for the highest score overall of 0.80. The mBERT model however was able to better identify arguments associated with the 4th and 6th verbal roots, by a mark of 0.04 and 0.06 respectively. This could be due to the fact that mBERT is better able to generalize, and as sentences including additional verbal roots occur far less frequently than those containing 1-3, XLM-R simply was not able to adapt. The top performing model overall, gal_ensp_XLM-R, also achieved top *f1* scores for the 1st-5th root sets and was the only model able to identify any argument of the 9th root set; receiving scores of 0.73, 0.68, 0.61, 0.62, 0.59, and 0.07 respectively. The Portuguese/Spanish pre-trained model ranks quite closely and outperforms the identification of 0th, 6th, and 7th roots. Given the linguistic similarity between Galician and Portuguese, it may be possible to achieve even better results if the Portuguese set is also pre-processed using the Verbal Indexing method in future work.

Looking at the Galician scoring overall, the ben-

| Model [e2c] | tot_avg | root | arg0 | arg1 | arg2 | arg3 | arg4 | argM |
|---|---|---|---|---|---|---|---|---|
| spa_mBERT [6] | 0.82 | **0.51** | **0.39** | 0.38 | 0.33 | 0.13 | 0.26 | 0.27 |
| spa_en_mBERT [6] | **0.81** | 0.49 | 0.37 | 0.37 | 0.32 | **0.09** | **0.22** | 0.26 |
| spa_pt_mBERT [7] | 0.82 | **0.44** | 0.37 | 0.38 | 0.32 | 0.20 | 0.27 | 0.29 |
| spa_enpt_mBERT [6] | **0.81** | 0.48 | **0.36** | **0.36** | **0.30** | 0.16 | 0.31 | 0.27 |
| spa_XLM-R [7] | 0.82 | **0.44** | **0.36** | 0.37 | 0.34 | 0.18 | 0.28 | **0.24** |
| spa_en_XLM-R [7] | **0.83** | 0.47 | 0.38 | **0.39** | 0.35 | **0.23** | **0.41** | 0.29 |
| spa_pt_XLM-R [7] | **0.83** | 0.49 | **0.39** | **0.39** | **0.36** | 0.19 | 0.34 | **0.30** |
| spa_enpt_XLM-R [6] | **0.83** | 0.49 | 0.37 | **0.39** | 0.35 | **0.09** | 0.29 | 0.28 |

Table 6: Final *f1* Scores: Spanish Models
‖ *e2c* – number of epochs trained until convergence; *tot_avg* – average *f1* score overall | *root* – average *f1* score identifying verbal roots | *arg0* – average *f1* score identifying argument 0 | *arg1* – average *f1* score identifying argument 1 | *arg2* – average *f1* score identifying argument 2 | *arg3* – average *f1* score identifying argument 3 | *arg4* – average *f1* score identifying argument 4 | *argM* – average *f1* score identifying argument M ‖

| Model [e2c] | r0 | r1 | r2 | r3 | r4 | r5 | r6 | r7 | r8 | r9 | r10 |
|---|---|---|---|---|---|---|---|---|---|---|---|
| spa_mBERT [6] | 0.528 | 0.529 | 0.470 | 0.439 | 0.430 | 0.360 | 0.229 | 0.184 | 0.055 | **0.033** | **0.000** |
| spa_en_mBERT [6] | **0.524** | **0.487** | 0.469 | **0.416** | 0.396 | 0.317 | 0.228 | 0.181 | 0.038 | 0.031 | **0.000** |
| spa_pt_mBERT [7] | 0.576 | 0.562 | 0.472 | 0.450 | 0.417 | 0.367 | 0.245 | 0.126 | **0.016** | 0.008 | **0.000** |
| spa_enpt_mBERT [6] | 0.543 | 0.540 | **0.459** | 0.417 | 0.423 | **0.376** | 0.194 | 0.131 | 0.033 | **0.000** | **0.000** |
| spa_XLM-R [7] | **0.612** | 0.548 | 0.491 | 0.431 | **0.365** | **0.288** | **0.169** | **0.110** | 0.029 | 0.004 | **0.000** |
| spa_en_XLM-R [7] | 0.606 | **0.573** | **0.499** | **0.469** | 0.447 | 0.324 | 0.245 | **0.306** | 0.070 | 0.010 | **0.000** |
| spa_pt_XLM-R [7] | 0.588 | **0.573** | 0.476 | 0.454 | **0.480** | 0.365 | **0.279** | 0.287 | 0.082 | 0.004 | **0.000** |
| spa_enpt_XLM-R [6] | 0.570 | 0.501 | 0.474 | 0.442 | 0.444 | 0.328 | 0.251 | 0.191 | **0.087** | 0.007 | **0.000** |

Table 7: Final *f1* Scores by Verb Index: Spanish Models
‖ *e2c* – number of epochs trained until convergence | *r[idx]* – average *f1* score identifying [idx]th root/arguments | r11-r14 scores not included as always = 0.00 | score of 0.00 signifies that no correct identifications of any kind were made by that model for the [idx]th root ‖

| | r0:arg0 | O | r0:root | O | r1:root | O | O | r1:arg1 | | O |
|---|---|---|---|---|---|---|---|---|---|---|
| **eng** | Soedade | López | explains | how | to educate on sexuality | | | | | . |
| **glg** | Soedade | López | explica | como | educar | para | a | sexualidade | | . |
| ***gal_mBERT** | r0:arg0 | O | r0:root | O | r1:root | O | O | r1:arg1 [sexual] | | O |
| ****gal_XLM-R** | r0:arg0 | O | r0:root | O | r1:root | O | O | r1:arg1 | | O |

Table 8: gal_mBERT and gal_XLM-R SRL Predictions: Galician Dataset Sentence #9
‖ *eng* – English gloss of glg | *glg* – Galician text | *gal_mBERT* – predicted roles according to gal_mBERT | *gal_XLM-R* – predicted roles according to gal_XLM-R | yellow – target label | green – correct prediction | red – incorrect prediction | orange – partially correct prediction [error in brackets] | *r[idx]:[label]* – link [label] to [idx]th root | *root* – token assigned as the verbal root | *arg0* – token assigned as argument 0 | *arg1* – token assigned as argument 1 | *O* – token assigned as not directly related to root | * – worst *f1* scoring model(s) | ** – best *f1* scoring model(s) ‖

efits of Verbal Indexing are clear; models not pre-trained on a dataset using this method were unable to identify arguments of the 6th root or later except for a single model, gal_en_XLM-R, achieving a score of 0.04 on the 7th root set. In comparison, the top performing model for the 7th root, gal_ptsp_XLM-R which does pre-train on the Verbal Indexing method, achieves a score of 0.28. This model was additionally able to identify arguments up through the 9th root set, 3 full root sets past those not pre-trained on the Verbal Indexing method.

Looking at the Spanish models, the benefits are less clear but still visible. Despite the difficulties seen in argument identification by label, spa_XLM-R was still able to tie the monolingual mBERT model due to its increased performance in root set identification in less-complex sentences; scoring +0.084, +0.019, and +0.021 in the identification of the 0th-2nd root sets over mBERT. This could be due to the fact that no Spanish model was pre-trained on the Verbal Indexing method, or perhaps that the benefits of Verbal Indexing lessen as the availability of training data increases; it is recommended that this be explored in future research.

## Conclusion

In this work, we presented the first-ever SRL dataset for the low-resource Galician language, thus establishing a baseline for future SRL research in Galician. We also introduced a new preprocessing method for SRL systems, verb indexing, which shows positive effects on the final performance of systems when analyzing more complex sentences. The best performing SRL model for Galician scored 0.74 *f1*, an increase of 0.06 over the Galician-only baseline model.

## Limitations

One limitation of this work is that the Galician dataset was created using a semi-automatic process, with only a small portion being manually validated. Ideally when developing corpora, manual annotations generated by native speakers are preferred as it ensures accuracy; a downside of this is the increased time, and likely financial, cost necessary to recruit a team of annotators (King, 2015). Utilizing an automatic process which is then partially manually validated is one method to reduce these costs. Due to external constraints, manual validation in this study was limited to a small portion of the dataset and performed by an individual with a non-native understanding of Galician.

Despite their linguistic similarities, the Portuguese dataset was also not pre-processed utilizing the Verbal Indexing method and it is recommended that this be explored in future work. Developing a reduced English dataset using this indexing method could also be quite helpful to other low-resource languages; as many pre-trained English models are available, it would be quite easy to fine-tune on a smaller English SRL dataset utilizing the indexing method, and then fine-tune the target low-resource language. The results reported here suggest this could have incredibly positive effects on the performance of SRL systems for low-resource languages, and potentially also higher-resource languages, and future related work should focus on exploring this.

## Ethics Statement

It is important to specify that this dataset was developed by an individual with a non-native understanding of Galician and that native speakers were unable to be included in this initial production of the dataset.

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
