# OpenReview forum: "BERTie Bott's Every Flavor Labels: A Tasty Introduction to Semantic Role Labeling for Galician"
_EMNLP/2023/Conference — EMNLP 2023 Main_

### Official Review · Reviewer_Dkyz · 2023-08-04

**Paper Topic And Main Contributions:** This model describes a  first SRL  da…
**Soundness:** 3

**Excitement:**

3: Ambivalent: It has merits (e.g., it reports state-of-the-art results, the idea is nice), but there are key weaknesses (e.g., it describes incremental work), and it can significantly benefit from another round of revision. However, I won't object to accepting it if my co-reviewers champion it.

**Reasons To Accept:**

Nice work. All choices for methodology are reasonable. This is a good findings paper and will help researchers interested in building similar corpora for other less-resourced languages.

What is missing is more discussion of why people from the ACL community should care, and what generalizable ideas have been uncovered as a result of the author's careful work.

**Reasons To Reject:**

Since subjective judgments are involved in the manual verification of some of the data, it would be a good idea to have inter-rater reliability numbers. Their absence is a weakness.

The limitations section mentions that the author (described as "an individual" is not a native speaker of Galician). This is not necessarily a problem, but it needs to be discussed in the body of the paper, not just mentioned in passing. The rater's actual language background should be revealed. If necessary, native speaker help should be added.

The most interesting lessons of the paper are not about the availability of a corpus for Galician, but the availability of a methodology that might be used for similar language situations. It is of interest if a small-scale effort by a non-native speaker can produce something useful. in order to make this lesson available, the author needs to indicate which aspects of the general language situation are important for the design of the corpus. This requires some material explaining the language situation. Despite having worked on relationships between Romance languages, this reviewer was not aware of the existence of Galician or its relationships with Spanish and Portuguese. If I am typical, ACL readers need some introductory material. Space could be found for this by reducing the amount of material about the various models used. It is not clear why the paper needs to report on so many models. If the author has hypotheses or opinions about what SHOULD work, and why, and results on what DID work, and why, they do not come through clearly in the paper. More selectivity is needed, and more focus on interesting general principles.

If I understand correctly, Galician has official status in a region of north western Spain, and is widely spoken, with a literary tradition, but also a period in which the language was suppressed by Franco's regime. The revival of the language is therefore new, possibly paralleling the situation of other languages that used to be banned.
These facts are potentially relevant to the wider applicability of the methods used for the study. For example, potential re-users of the methods need to know that there is a Wikipedia and that there is just one active newspaper, published in Galician and Spanish.

It looks to me that the results here are suitable for a Findings paper, but there is not much to excite a general ACL audience. People with strong interests in less resourced languages will be interested.

**Reproducibility:**

3: Could reproduce the results with some difficulty. The settings of parameters are underspecified or subjectively determined; the training/evaluation data are not widely available.

**Reviewer Confidence:**

4: Quite sure. I tried to check the important points carefully. It's unlikely, though conceivable, that I missed something that should affect my ratings.

**Typos Grammar Style And Presentation Improvements:**

There is no good reason for the Harry Potter reference in the title. An informative title is hugely preferable.

---

> ### Author Rebuttal · Authors · 2023-08-29
>
> We genuinely appreciate your thoughtful feedback, and we are committed to addressing these points to improve the overall quality and relevance of our paper.
>
> ´´It looks to me that the results here are suitable for a Findings paper, but there is not much to excite a general ACL audience. People with strong interests in less-resourced languages will be interested.''
> We value your perspective on the potential interest of the ACL audience. While our primary focus is on providing insights for researchers working with less-resourced languages, we are also proposing a broader methodology as you mentioned. Your insight into the broader applicability of our methodology aligns with our intention. We aim to highlight not just the creation of a Galician corpus but also the potential relevance of our approach to languages facing similar resource challenges. In response to your suggestion, we will provide more explicit guidance on adapting our methodology to address the general language situation and challenges, offering a framework for researchers working with languages in comparable circumstances. We will take this feedback into account and work to enhance the discussion of general principles and methodology that could resonate with a wider audience within the ACL community.

---

### Official Review · Reviewer_Swd1 · 2023-08-05

**Soundness:** 3

**Excitement:**

4: Strong: This paper deepens the understanding of some phenomenon or lowers the barriers to an existing research direction.

**Missing References:**

- Conia, Simone, Edoardo Barba, Alessandro Scirè, and Roberto Navigli. ‘Semantic Role Labeling Meets Definition Modeling: Using Natural Language to Describe Predicate-Argument Structures’, 2022. https://doi.org/10.48550/ARXIV.2212.01094.
- He, Shexia, Zuchao Li, Hai Zhao, and Hongxiao Bai. ‘Syntax for Semantic Role Labeling, To Be, Or Not To Be’, Proceedings of the 56th Annual Meeting of the Association for Computational Linguistics (Long Papers), pages 2061–2071. Melbourne, Australia, July 15 - 20, 2018.
- Màrquez, Lluís, Xavier Carreras, Kenneth C. Litkowski, and Suzanne Stevenson. ‘Semantic Role Labeling: An Introduction to the Special Issue’. Computational Linguistics 34, no. 2 (June 2008): 145–59. https://doi.org/10.1162/coli.2008.34.2.145.

**Paper Topic And Main Contributions:**

The authors created a dataset of Semantic Role Labeling for Galician, a low-resource language. They chose to focus on verbal roots and their arguments in order to identify semantic roles (2.0.1). Then they procede to prune the arguments and to attribute a specific label to specific kinds of arguments. For Spanish, they prepare the existing data for the verbal indexing step (2.0.2). They also evaluated the impact of verbal indexing as pre-processing for this task on Galician and Spanish.

**Questions For The Authors:**

- Question A: What are the grounds for choosing Spanish as a source for Galician? It is also a Romance language and their localisations are close, but having a few pieces of information about vocabulary and grammar similarities would help understand the method. A majority of speakers are probably bilingual, which may have an impact on the annotation.
- Question B: Just one test or cross-validation? It could change the differences between the setups.
- Question C: What quantity of data do you have? How did you divide your data into train, dev and test?
- Question D: Are there more examples of different treatments by WordPiece and SentencePiece?

**Reasons To Accept:**

- There was no such dataset for Galician. This work provides a new resource for a low-resource language and tries methods for this kind of language.
- The authors have a sound approach: they evaluated their pre-processing method on a second language with more resources and speakers.
- Comparing the the models' ability to find up to 10 semantic label in a sentence is an interesting and valuable evaluation.
- Comparing WordPiece and SentencePiece for this task is valuable feedback for the community.

**Reasons To Reject:**

- Not much has been explained about disambiguation of the annotation, although it is an important topic.
- As far as I understand, the authors did not look into cross-validating their evaluations. A difference in score can be nuanced by changing train, dev and test distributions.
- Verbal indexing seems to be a new name for what is common practice in the literature. The structure given by a predicate and its arguments is already the foundation of semantic role labeling (see missing references for examples).

**Reproducibility:**

3: Could reproduce the results with some difficulty. The settings of parameters are underspecified or subjectively determined; the training/evaluation data are not widely available.

**Reviewer Confidence:**

2: Willing to defend my evaluation, but it is fairly likely that I missed some details, didn't understand some central points, or can't be sure about the novelty of the work.

---

> ### Author Rebuttal · Authors · 2023-08-29
>
> We thank the reviewer for their insightful feedback and suggestions. Below, we offer responses to the posed inquiries:
> A. Few languages are categorized as high-resource for this task. Spanish stands out not only due to its abundant SRL data and published results that serve as a baseline but also because we had annotators who could speak both Spanish and Galician.
>
> B. Although this experiment did not involve cross-validation through alterations in train/test/dev groups, the inclusion of numerous models was deliberate. This approach provides a broader perspective on the results and accommodates potential variations.
>
> C. The Galician dataset comprises 3,987 training sentences and 998 test sentences, encompassing sentences with up to 13 verbal roots. In contrast, the Spanish dataset includes 14,329 training sentences, 1,655 development sentences, and 1,725 test sentences. Each of these sentences hosts up to 16 verbs. The train/test/dev split for Spanish adheres to the CoNLL release. For Galician, an 80/20 split was employed, with sentence allocation to sets being random yet with an endeavour to ensure a balanced distribution of simple (0-4 verbal roots) and more intricate (5+ verbal roots) sentences across sets, guided by insights gained from verbal indexing.
>
> D. A customary approach involves tracking word positions within a sentence. After reviewing the provided references, it seems this might align with your inquiry. Verbal indexing is specific to counting verbs within a sentence and assigning them indices. It does not pertain to sentence structure established by predicates and their arguments.
>
> We appreciate the feedback and will add more statistics about our dataset to the paper.

---

### Official Review · Reviewer_9UHD · 2023-08-11

**Soundness:** 2

**Excitement:**

3: Ambivalent: It has merits (e.g., it reports state-of-the-art results, the idea is nice), but there are key weaknesses (e.g., it describes incremental work), and it can significantly benefit from another round of revision. However, I won't object to accepting it if my co-reviewers champion it.

**Missing References:**

(see above Questions)

Lonneke 2011
Conia 2021

**Paper Topic And Main Contributions:**

This paper claims two contributions: a new Galician SRL dataset and a new preprocessing method, verbal indexing. There are indeed not many Galician resources so this would benefit the community. On the other hand the verbal indexing method does not seem to me to be new.

**Questions For The Authors:**

The verbal indexing looks very similar to to direct mapping in Lonneke, 2011. That approach was criticized in Akbik, 2015 (which you do cite) which showed filtered projection to work better. Can you explain the distinction better between Lonneke and you, and more clearly defend the novelty?

I don't see a clear description how you matched the verbal lemma in Galician to its English PropBank roleset, which is an important part of the methodology. In any language there will be cases where you can not easily match the target language verb to its English roleset.  For example, phrasal verbs in English can map to a single verb in the target language and so on.  Please augment the paper by describing your matching algorithm.

There are other SoTA results on multilingual SRL that ought to be added to the experiments. For example Conia, 2021 (Unifying Cross-Lingual Semantic Role Labeling with Heterogeneous Linguistic Resources) shows Spanish SoTA at 86.3, higher than this paper's 83 (interestingly, also using XLM-R). Please do additional lit review (or else explain why this work does not apply, as I believe it does) and expand your experimental section accordingly




**Reasons To Accept:**

More work in low resource languages is always good, we need Galician resources.

**Reasons To Reject:**

-The contribution of the verbal indexing method does not seem to me to be new
-There are some clarifications needed in the methodology section, see below in Questions
-The experiments section does not appear to compare to SoTA, see below in Questions

**Reproducibility:**

3: Could reproduce the results with some difficulty. The settings of parameters are underspecified or subjectively determined; the training/evaluation data are not widely available.

**Reviewer Confidence:**

4: Quite sure. I tried to check the important points carefully. It's unlikely, though conceivable, that I missed something that should affect my ratings.

**Typos Grammar Style And Presentation Improvements:**

Overall the presentation is fairly clear.  I did find the captions extremely challenging to read in the way that they were formatted - I suggest fewer details and more description of what the takeaway point of the figure / table is.

---

> ### Author Rebuttal · Authors · 2023-08-29
>
> We value the feedback from the reviewer and aim to address the concern raised by R2 regarding the lack of innovation in our model's verb indexing approach.
>
> Our approach to verb indexing differs from a direct mapping technique. Essentially, verb indexing involves tallying the number of verbs present in a given sentence. This is done to enhance the model's understanding of sentence complexity and aid in the initial identification of verbs within the sentence. It's important to note that this method does not play a role in assigning semantic roles or rolesets, unlike the direct projection method. The purpose of verbal indexing is to count all verbs within a sentence, arranging them based on their position in the sentence. Importantly, this indexing process does not impact the PropBank label associated with each verb. For instance, even if a verb with the PropBank label "run.01" appears twice in a single sentence, each instance would be assigned a distinct index through the verbal indexing approach, as the index is directly linked to the verb's position in the sentence. We will add all missing references and add a paragraph about the differences between verb indexing and direct mapping.

---

### Official Review · Reviewer_x4WB · 2023-08-22

**Soundness:** 4

**Excitement:**

4: Strong: This paper deepens the understanding of some phenomenon or lowers the barriers to an existing research direction.

**Paper Topic And Main Contributions:**

The paper, "BERTie Bott’s Every Flavor Labels: A Tasty Introduction to Semantic Role Labeling for Galician," delves into the domain of Semantic Role Labeling (SRL) with a specific focus on the Galician language. SRL is a computational linguistic task that identifies the semantic roles of words in a sentence concerning a particular noun or verb. The study emphasizes the importance of SRL in relation to verbal predicates.

Main Contributions:
1 Creation of a Galician SRL Dataset: The authors successfully leveraged existing resources, such as the UD Galician-CTG and UD Galician-TreeGal treebanks, to produce the first-ever dataset for training SRL systems in Galician. This dataset consists of 3,987 training sentences and 998 test sentences.
2 Introduction of "Verbal Indexing": A novel pre-processing method named "verbal indexing" was introduced. This technique aids in semantically parsing complex sentences, enhancing the accuracy and efficiency of the SRL process.
3 Transfer Learning Experiments: The paper employed transfer learning to test the newly created resource and the verbal indexing method. A total of 24 models were developed and tested, with 16 tailored for Galician and 8 for Spanish.

**Questions For The Authors:**

Question A: Can you provide more details on the diversity of the Galician SRL dataset in terms of genres, styles, and sources? How do you ensure that it captures a broad spectrum of the Galician language?

Question B: How does the "verbal indexing" method perform when applied to languages with significantly different linguistic structures than Galician or Spanish? Have you considered testing it on non-Romance languages?

Question C: In your experiments, where did the models particularly struggle or fail? A deeper analysis of these challenges could provide insights for future improvements.

**Reasons To Accept:**

1 Addressing a Low-Resource Language: One of the paper's primary strengths is its focus on the Galician language, a low-resource language in the NLP domain. By introducing resources and methodologies for such a language, the paper contributes to the diversification and inclusivity of linguistic research in NLP.
2 Creation of a Novel Dataset: The authors have successfully developed the first-ever dataset for training SRL systems in Galician. This dataset fills a significant gap in the NLP resources available for the language and provides a foundation for future research in this area.
3 Introduction of "Verbal Indexing": The novel pre-processing method, "verbal indexing," introduced in the paper, offers a fresh perspective on semantically parsing complex sentences. This technique can potentially be applied to other languages and tasks, making it a valuable contribution to the field.
4 Use of Transfer Learning: The application of transfer learning to test the new resource and the verbal indexing method showcases a modern approach to NLP tasks, emphasizing the paper's relevance to current research trends.

**Reasons To Reject:**

1 Dataset Size and Diversity: While the creation of a Galician SRL dataset is commendable, its size might be considered relatively small for deep learning models. The diversity of the dataset in terms of genres, styles, and sources could also be a concern, as it might not capture the full linguistic richness of the Galician language.
2 Generalizability of "Verbal Indexing": The novel "verbal indexing" method is introduced and tested primarily on Galician and Spanish. Its effectiveness and applicability to other languages, especially those with different linguistic structures, remain unexplored.
3 Detailed Analysis of Failures: While the paper presents the successes of its methods, a deeper dive into where the models failed or struggled would provide a more balanced view and insights for future improvements.

**Reproducibility:**

4: Could mostly reproduce the results, but there may be some variation because of sample variance or minor variations in their interpretation of the protocol or method.

**Reviewer Confidence:**

4: Quite sure. I tried to check the important points carefully. It's unlikely, though conceivable, that I missed something that should affect my ratings.

---

> ### Author Rebuttal · Authors · 2023-08-29
>
> We would like to thank the reviewer for providing feedback, and insightful suggestions. Below, we respond to the questions asked.
>
>  A. The Galician dataset includes technical sentences of the medical, sociological, ecological, economic, and legal domains that have been pulled from the both Galician Technical and XIADA Corpus by the Galician-CTG and Galician-TreeGal treebanks. We will add that to the paper.
>
> B.  It would be interesting to branch out of romance languages to see if verbal indexing has the same effect on complex sentences in other language families and we do hope to do so in the future, however, this study was meant as a beginning and its main focus was increasing available resources for low-resource Galician. The inclusion of Spanish was a lesser focus and was mainly used as a method of comparing the results against an established baseline.
>
> C. mBERT models (using WordPiece tokenization) mainly struggled throughout with correctly tokenizing individual words, often breaking tokens down too far and assigning roles to only portions of a single token.
>
> XLM-R models (using SentencePiece tokenization) did not have this issue and instead mainly struggled with correctly identifying verbs/arguments in more complex sentences; for example in a sentence containing 6 verbs, a model may have only correctly identified the first four verbs and their arguments. The final performance was increased when pre-trained using the verbal indexing method, but the XLM-R models still struggled more with this than the mBERT models.

---

### Meta-Review · Area_Chair_8WXX · 2023-09-19

**Recommendation:** 2

**Metareview:**

This submission introduces a training dataset for Galician Semantic Role Labeling, and a pre-processing method called verb indexing, to improve the accuracy of parsing complex sentences. The proposed method has been also evaluated on Spanish to measure its impact.
After author rebuttal, the reviewers discussed it, and they agreed that a new resource for Galician is valid and worthy for the community. However, they also point out the need to improve the evaluation comparison with recent multilingual SoTA SRL works that provide better results (not included in the submission). Moreover, the paper would also benefit from a better explanation of the novelty of their proposed verbal indexing method.

---

### Decision · Program_Chairs · 2023-10-07

**Decision:**

Accept-Main

**Comment:**

This submission introduces a training dataset for Galician Semantic Role Labeling, and a pre-processing method called verb indexing, to improve the accuracy of parsing complex sentences. The proposed method has been also evaluated on Spanish to measure its impact.
After author rebuttal, the reviewers discussed it, and they agreed that a new resource for Galician is valid and worthy for the community. However, they also point out the need to improve the evaluation comparison with recent multilingual SoTA SRL works that provide better results (not included in the submission). Moreover, the paper would also benefit from a better explanation of the novelty of their proposed verbal indexing method.